# Lignin-Furanic Rigid Foams: Enhanced Methylene Blue Removal Capacity, Recyclability, and Flame Retardancy

**DOI:** 10.3390/polym16233315

**Published:** 2024-11-27

**Authors:** Hugo Duarte, João Brás, El Mokhtar Saoudi Hassani, María José Aliaño-Gonzalez, Solange Magalhães, Luís Alves, Artur J. M. Valente, Alireza Eivazi, Magnus Norgren, Anabela Romano, Bruno Medronho

**Affiliations:** 1MED—Mediterranean Institute for Agriculture, Environment and Development, CHANGE–Global Change and Sustainability Institute, Faculdade de Ciências e Tecnologia, Universidade do Algarve, Campus de Gambelas, 8005-139 Faro, Portugalmariajose.aliano@gm.uca.es (M.J.A.-G.); aromano@ualg.pt (A.R.); bfmedronho@ualg.pt (B.M.); 2Laboratory of Engineering Electrochemistry, Modelling, and Environment, Department of Chemistry, Faculty of Sciences Dhar Mahraz, Sidi Mohamed Ben Abdellah University, Fez 30050, Morocco; 3Analytical Chemistry Department, University of Cádiz, 11510 Puerto Real, Spain; 4University of Coimbra, CERES, Department of Chemical Engineering, Pólo II–R. Silvio Lima, 3030-790 Coimbra, Portugal; solangemagalhaes@eq.uc.pt (S.M.);; 5CQC-IMS, Department of Chemistry, University of Coimbra, 3004-535 Coimbra, Portugal; avalente@ci.uc.pt; 6Surface and Colloid Engineering, FSCN Research Center, Mid Sweden University, SE-851 70 Sundsvall, Sweden; alireza.eivazi@miun.se (A.E.); magnus.norgren@miun.se (M.N.)

**Keywords:** lignin, polyphenols, tannins, water treatment, biobased materials, foams, methylene blue

## Abstract

Worldwide, populations face issues related to water and energy consumption. Water scarcity has intensified globally, particularly in arid and semiarid regions. Projections indicate that by 2030, global water demand will rise by 50%, leading to critical shortages, further intensified by the impacts of climate change. Moreover, wastewater treatment needs further development, given the presence of persistent organic pollutants, such as dyes and pharmaceuticals. In addition, the continuous increase in energy demand and rising prices directly impact households and businesses, highlighting the importance of energy savings through effective building insulation. In this regard, tannin-furanic foams are recognized as promising sustainable foams due to their fire resistance, low thermal conductivity, and high water and chemical stability. In this study, tannin and lignin rigid foams were explored not only for their traditional applications but also as versatile materials suitable for wastewater treatment. Furthermore, a systematic approach demonstrates the complete replacement of the tannin-furan foam phenol source with two lignins that mainly differ in molecular weight and pH, as well as how these parameters affect the rigid foam structure and methylene blue (MB) removal capacity. Alkali-lignin-based foams exhibited notable MB adsorption capacity (220 mg g^−1^), with kinetic and equilibrium data analysis suggesting a multilayer adsorption process. The prepared foams demonstrated the ability to be recycled for at least five adsorption-desorption cycles and exhibited effective flame retardant properties. When exposed to a butane flame for 5 min, the foams did not release smoke or ignite, nor did they contribute to flame propagation, with the red glow dissipating only 20 s after flame exposure.

## 1. Introduction

To mitigate human hazards and improve environmental health, there is an increasing need to efficiently remove pollutants from wastewater effluents before their discharge into the environment. The control of water contamination has become a great challenge due to toxic heavy metals, organic dyes, and other persistent organic pollution from several industries, such as textile, cosmetic, paper, and pharmaceutical [1,2]. With only 1% of accessible water for human and industrial needs, water scarcity has become a significant issue. The 2007 report by the United Nations FAO identifies that water stress could affect two-thirds of the world’s population. To alleviate such stress, wastewater remediation could be a suitable solution by recovering water from industrial activities [3]. In this regard, several biological, chemical, and physical methods have been developed to minimize their spread and release into aquifers [1,2,4]. Among these, adsorption is one of the most attractive methods in wastewater treatment due to its cost effectiveness, facile operation, and broad applicability in contrast to other existing technologies [4].

Ideally, to minimize the environmental impact of adsorption processes, an adsorbent should be inert, effective in removing the target molecule(s), made from renewable and abundant materials, and require minimal processing before use [5]. In this regard, polyphenol-based rigid foams are materials mainly obtained from natural products, known for their fire resistance, adsorption performance, and easy preparation [6,7,8]. Among the different polyphenols, tannin-based biosorbents are known to have a natural affinity toward dyes, heavy metals, pharmaceutical agents, and surfactants from contaminated waters [7,9]. Moreover, these systems can selectively accumulate important precious metals from aqueous streams or enhance their adsorption capacity for various compounds when functionalized [5,10,11,12].

Another very appealing polyphenol often used in wastewater treatment is lignin [1,13]. Lignin is the second most abundant renewable biomolecule on Earth and is regarded as a sustainable aromatic feedstock that can compete and substitute for petroleum-based phenol [14]. Nowadays, most of the available lignin is obtained from the chemical pulping of wood using the kraft and the sulfite cooking processes [15]. The pulping process from which lignin is obtained essentially dictates the content and type of functional groups, such as phenolic, carboxyl, hydroxyl, and sulfonate. With its high carbon content, biodegradability, antioxidant activity, thermal stability, and advantageous stiffness, lignin is seen as an attractive material for several applications, including drug delivery systems, emulsion stabilization, rechargeable batteries, and carbon-based materials [16,17,18,19]. 

Apart from lignin, lignocellulosic biomass is the main feedstock for furfural, the main precursor to produce furfuryl alcohol [20]. The reaction between furfuryl alcohol and natural polyphenolic sources, such as lignin and tannins, has been sought to produce biobased materials. Tannin-furan (TF) foams have been studied as suitable biobased alternatives, mainly for the replacement of fossil-based materials as insulators or flame retardants [21,22,23,24]. This is particularly relevant for energy savings purposes, which is becoming a global priority. Effective building insulation emerges as an important solution, offering a substantial reduction in energy consumption for heating and cooling as the energy needed to achieve thermal comfort accounts for approximately 50% of the total residence consumption [24]. By improving the thermal efficiency of buildings, insulation helps to maintain consistent indoor temperatures, thereby reducing the need for excessive energy use. This not only lowers energy bills but also contributes to a more sustainable and environmentally friendly approach to energy consumption. Investing in high-quality insulation is, therefore, essential for mitigating the economic impact of rising energy prices and enhancing overall energy efficiency. 

Polyphenol-based rigid foams have been prepared via different procedures and extensively tested, demonstrating desirable characteristics, such as good mechanical strength and flame retardant properties [25,26]. Tannins and lignin have been incorporated into foam formulations in distinct ways: as substitutes for fossil-derived phenol, to replace formaldehyde, or to enhance the foam’s mechanical and thermal properties [22,27,28]. However, a gap in the literature shows that no comparisons have been made regarding how the addition of tannin or different lignins to furanic foams affects their dye removal capacity. In the present work, novel formaldehyde-free tannin- and lignin-furan-based foams with several desirable characteristics were prepared. Using the TF foam as standard, kraft or alkali lignin was progressively added to investigate how lignin inclusion in the foam formulation would affect the adsorption performance of each foam [24]. The biobased foams were successfully characterized and tested for the removal of a model dye (i.e., methylene blue) from aqueous media, as the adsorption process was studied in terms of the influence of dye concentration, time, and pH. Several kinetic and isotherm models were applied to understand the mechanism of adsorption and the rate-limiting step of the process. The foam recyclability and reusability were also evaluated. This study also aims to present formaldehyde-free rigid foams as a multipurpose material to be used as a recyclable adsorbent, with a second life application as a flame retardant material. 

## 2. Materials and Methods

### 2.1. Materials

Phenol (>99.5%), formaldehyde (37%), pentane (99%), p-toluenesulfonic acid (≥98.5%), sodium bicarbonate (>99%), sodium chloride (>99.5%), potassium chloride (>99.5%), methylene blue, hydrochloric acid (37%), and “alkali” lignin (Mw = 10 kDa, pH 10.5, sulfur < 3.6 wt%) were purchased from Sigma Aldrich (St. Louis, MO, USA), while “kraft” lignin (UPM BioPiva^TM^ 100, Mw = 5 kDa, pH 2.5–4.5, sulfur 0.5–3 wt%) was acquired from UPM Biochemicals. The “Opera fruity” (condensed tannin extract from red fruits with a total polyphenol content > 65 wt%) was obtained from Proenol, SA (Porto, Portugal). Ethanol (96%) was acquired from Aga (Lisbon, Portugal), sodium carbonate anhydrous (99.7%) from José Manuel Gomes dos Santos, Lda (Lisbon, Portugal), and sodium hydroxide (≥98%), disodium hydrogen phosphate (>99%), and potassium dihydrogen phosphate (>99%) from Panreac (Barcelona, Spain). 

### 2.2. Preparation of the Tannin and Lignin-Based Rigid Foams

The foams were prepared at room temperature, following a standard procedure with slight modifications (see Table 1 for details) consisting of mixing approximately 2.5 g of polyphenol (i.e., tannin, lignin, or different ratios of both) with distilled water (i.e., 1 mL) inside a flexible plastic mold of 2 cm^3^ with the help of a glass rod, until a homogenous suspension was obtained. Then, further lignin or tannin was added in increasing amounts (i.e., 0.2, 1.25, and 2.5 g) to the suspension and mixed until homogeneity. Furfuryl alcohol (i.e., 1.4 mL) was added, and the solution was mixed until a homogenous mixture was obtained. After this, pentane was added as the blowing agent. p-Toluenosulfonic acid (p-TSA) (ca. 0.9 g) was then added in crystal form and mixed until total dissolution was observed. Once homogenized, the mixture was left to settle for approximately 20 to 30 s as the foaming reaction initiated and evolved. After the reaction completion, the “fresh” foam was left resting for approximately 10 min to ensure no morphological changes. The foams were then removed from the plastic mold and cured overnight (for 12 h) in an oven at 60 °C [29]. It is important to note that given the success in totally replacing the tannin with lignin, the foams chosen for further characterization were those with only one phenol source (highlighted in Table 1). As mentioned above, the tannin-furan foam (TF) served as the control, as it was the starting foam in which tannin was progressively replaced by lignin. Nevertheless, preliminary assays demonstrated that the addition of tannin does not significantly improve dye removal. 

#### Foam Washing

Before usage in adsorption kinetics experiments, foams were weighed, measured, and cut. The foams, of approximately 1 g, were immersed in 100 mL distilled water and left overnight to rinse any chemical that might not have reacted. The water was changed after 12 h, and the process repeated once. After the second rising, the foams were dried in an oven at 60 °C for 12 h. UV-vis was used to follow the efficiency of rinsing. The analyzed spectra were identical for all the effluents from the washed foams, presenting an absorption maximum of approximately 220–290 nm. Identified maxima adsorption is mainly related to unreacted tannin or lignin and residual furfuryl alcohol (<1 wt% (*w*/*w*)) [30]. Foams were stored in a desiccator until further analysis.

### 2.3. Foam Characterization Methodology

IR analyses were performed using a Bruker Tensor 27 (Billerica, MA, USA). Specimens were prepared from the different dried foam powders and pressed with KBr, forming pellets, and analyzed from 600 to 4000 cm^−1^ with 25 scans and a 4 cm^−1^ interval at room temperature [31]. 

Field emission scanning electron microscopy (FE-SEM) imaging of the samples was carried out using a TESCAN (Brno, Czechia) MAIA3 electron microscope in secondary electrons mode. The accelerating voltage was 3 kV, and the work distance was set to 8 mm. Before image acquisition, the samples were coated with 6 nm Iridium using a Quorum Q150T ES [32]. 

X-ray diffraction (XRD) was conducted at room temperature using a Bruker D2 Phaser diffractometer with Cu Kα radiation (wavelength 1.54 Å) at 30 kV and 10 mA, in θ–2θ geometry. The increment was fixed at 0.02° [32].

Fire retardancy and self-extinguishing properties of the prepared foams were assessed by a homemade flame test and thermogravimetric analyses (TGA). TGA was performed using a simultaneous DSC-TGA thermal analyzer (TA Instrument SDT Q600, New Castle, DE, USA). The samples were heated from room temperature up to 1200 °C, at a rate of 10 °C min^−1^, under a nitrogen atmosphere. Flame retardancy tests were executed by exposing a foam (2 cm^3^) to a Bunsen burner with a butane flame (over 1500 °C) for 5 min. After cutting the gas flow, samples continued to be recorded to estimate how long it would take for a flame or glow to dissipate [24].

### 2.4. Adsorption-Desorption Experiments

#### 2.4.1. Adsorption Kinetics

The adsorption batch experiments were carried out in 250 mL Erlenmeyer flasks at room temperature, where 100 mg of grounded foam was added per 100 mL of methylene blue buffered solution (3 × 10^−5^ M, pH 7.2 ± 0.2). Solutions were placed in a Edmund Bühler, (Bodelshausen, Germany) rotary mixer at 125 rpm, and aliquot samples were periodically taken, at least as duplicates. To make sure no foam powder fragments interfere with the UV-vis measurements, all aliquots were centrifuged for 2 min at 12,000 rpm (Mikro Hettic 200) before measuring the absorbance in a UV-vis spectrometer (T70+ UV/VIS Spectrometer, PG instruments Ltd.) at 664 nm (i.e., methylene blue’s maximum absorption). The quantity of adsorbed dye (*q_e_*) per gram of adsorbate in equilibrium conditions (mg g^−1^) was calculated from Equation (1) [4,33],
(1)$$q_{e}=\frac{\left(C_{0}-C_{e}\right)V}{W}$$

Here, *C*_0_ (mg L^−1^) is the initial concentration of dye in solution, *C_e_* (mg L^−1^) is the equilibrium concentration of adsorbate remaining in solution, *V* (L) is the volume of the solution, and *W*(g) is the adsorbent’s mass.

The removal efficiency (%*R*) was assessed using Equation (2),
(2)$$\%R=\frac{C_{0}-C}{C_{0}}\cdot 100=\frac{C_{ads}}{C_{0}}\cdot 100$$
where *C_ads_* (mg L^−1^) is the concentration of adsorbed dye and *C* (mg L^−1^) is the equilibrium concentration of the dye in solution [33].

The obtained adsorption kinetic data at neutral pH for 5, 10, 20, and 50 mg L^−1^ of MB were also modeled using pseudo-first-order (PFO) and pseudo-second-order (PSO) equations, respectively, as follows,
(3)$$q_{t}=q_{e}(1-e^{-k_{1}t})$$
(4)$$q_{t}=\frac{k_{2}q_{e}^{2}t}{1+k_{2}q_{e}t}$$
where *q_t_* (mg g^−1^) is the amount of adsorbate at time *t* (min), *k*_1_ (min^−1^) is the rate constant for the PFO, and *k*_2_ (g mg^−1^ min^−1^) for the PSO [33]. Data were represented as the averages calculated from three independent measurements and corresponding standard deviation.

#### 2.4.2. Adsorption in Equilibrium

For the evaluation of the isotherms, equilibrium batch experiments were performed for 24 h in 100 mL Erlenmeyer flasks loaded with 10 mL of 1, 2, 5, 10, 20, 30, and 50 mg L^−1^ of MB diluted with a buffer to control the pH. The AF foam was further tested in 75, 100, 150, 200, and 250 mg L^−1^ MB solution. Before measuring the absorbance of the solution, samples were centrifuged and diluted if necessary. Each experiment was performed in duplicate. The experimental data were fitted using the equations of Langmuir, Freundlich, Brunauer–Emmet–Teller (BET), and Dubinin–Radushkevich (D-R). The Langmuir isotherm model (Equation (5)) assumes a monolayer distribution of the adsorbate, a homogeneous distribution of the adsorption sites, a constant energy of adsorption, and neglects interactions between adsorbate molecules. It also assumes that the maximum adsorption corresponds to a monosaturated layer of adsorbate molecules on the surface of the adsorbent [34,35].
(5)$$q_{e}=\frac{q_{m}K_{L}C_{e}}{1+K_{L}C_{e}}$$

Here and for the following equations, *q_m_* (mg g^−1^) is the maximum adsorption capacity per unit weight of adsorbent and *K_L_* (L mg^−1^) is the Langmuir constant. 

The Freundlich model (Equation (6)) is not restricted to monolayer formation and considers the possibility of multilayer adsorption. It represents a nonlinear adsorption process considering the heterogeneity of the surface of the adsorbent [34,35,36].
(6)$$qe=K_{F}{C_{e}}^{1/n}$$

*K_F_* (L^1/n^ mg^1−1/n^ g^−1^) is the Freundlich constant, and *n* is the Freundlich exponent, which is dimensionless. The BET (Equation (7)) model is a theoretical multilayer physical adsorption formalism, assuming homogenous adsorption where the adsorption energy of the first layer differs from the other layers. The model also takes a pseudo-steady state into account, a dynamic equilibrium where the rates of adsorption and desorption are equivalent [34,37].
(7)$$q_{e}=\frac{q_{m}K_{L}C_{e}}{\left(1-K_{S}C_{e}\right)[1+\left(K_{L}-K_{S}\right)C_{e}]}$$

*K_L_* (L mg^−1^) is the monolayer adsorption equilibrium constant, and *K_S_* is the multilayer adsorption equilibrium constant (L mg^−1^). Note that when *K_L_* = 0, the BET isotherm is reduced to the Langmuir model, representing the case of a monolayer adsorption process, and *K_L_* becomes the Langmuir constant.

Lastly, the D-R isotherm model is usually used to express an adsorption mechanism with a Gaussian energy distribution into heterogeneous surfaces [34,36].
(8)$$q_{e}=q_{m}-e^{-K_{D}\varepsilon^{2}}$$

*K_D_* (mol^2^ kJ^−2^) is the D-R model constant, and *ε* (kJ mol^−1^) is the adsorption potential which can be calculated using Equation (9) [34],
(9)$$\varepsilon =RTln\frac{C_{s}}{C_{e}}$$
where *C_s_* (mg L^−1^) is the solubility of the adsorbate and *C_e_* (mg L^−1^) is the equilibrium concentration of adsorbate remaining in solution. The mean free energy, *E* (kJ mol^−1^), can be calculated using Equation (10),
(10)$$E=\frac{1}{\sqrt{2K_{D}}}$$

This model is commonly applied for the determination of a mainly physical (*E* < 8 kJ mol^−1^) or chemical (8 < *E* < 16 kJ mol^−1^) adsorption process. 

#### 2.4.3. Adsorption in Acidic and Alkaline pH Conditions

To study the effect of pH in MB adsorption, the point of zero charge was first calculated. Briefly, approximately 0.3 g of powdered foam was added to a solution of fixed pH and measured again after 24 h. After this, equilibrium tests were performed as previously described at different pH values for a fixed MB concentration of 50 mg L^−1^. The AF foam was then tested at pH 7 and pH 10 up to 250 mg L^−1^ of MB [4,38].

#### 2.4.4. Desorption Tests

To evaluate the reusability and recyclability of the most promising foams, the remaining solution after the adsorption experiments was discarded, and the material containing adsorbed MB was dried overnight in an oven at 60 °C. After drying, a certain amount of foam was weighed, and ethanol, acetone, or 0.1 M HCl was added, keeping a ratio of 100 mg of foam per 100 mL of solution. The mixture was kept under agitation for 24 h at 125 rpm, and the absorbance of the solution was measured at 664 nm. The process was repeated five times and performed in duplicate [39].

## 3. Results and Discussion

### 3.1. Rigid Foam Synthesis and Adsorption Dependence from Lignin Type and Concentration

Tannin-furan (TF) foams and lignin composites are well known as adsorbents [4,11,40,41]. The foams produced result from a crosslink between the phenol source (tannin or lignin), furfuryl alcohol, and p-TSA acid, originating a porous and solid material (Figure 1a,b) [23].

It is of notice the difference between kraft and alkali-based foams; as the alkali foam does not self-blows, a denser rigid foam is thus obtained (Figure 1c). In fact, each additive showed marked differences in the foam capacity to self-blow, which was reflected in their density. The TF foam presented a density of 0.325 gcm^−3^, while the kraft lignin foam (KF) density was 0.285 gcm^−3^. Similar tannin or lignin-based foams were obtained with comparable densities, ranging from 0.08 gcm^−3^ to 0.32 gcm^−3^; however, their composition varied slightly by using diethyl ether as a blowing agent, a different tannin, and additives as surfactants, glyoxal or boric acid in addition to p-TSA [22,26]. As expected, due to its inability to self-blow and obtain a more compact foam, the alkali lignin foam (AF) density was the highest (0.900 gcm^−3^). This remarkable increase in the foam density is promoted by the increase in lignin content, increasing the viscosity of the initial suspension, which ultimately results in a less porous and denser foam [25]. Moreover, the alkali lignin possesses an alkaline pH (i.e., pH 10.3 at 3 wt%), causing a greater demand for acid to trigger the polymerization reaction. Therefore, increasing the amount of p-TSA acid from 0.9 g to 1.8 g allowed the foam to self-blow. 

Scanning electron micrographs of the prepared rigid foams revealed mostly smooth, flat surfaces, in which small particles contribute to differences in roughness and particle size according to the phenol source (Figure 2a–c). Across the literature, rigid foams are often seen through SEM as highly porous structures, which was not the present case given the processing of the foam [24,26,29]. Note that due to its application in a grounded form, all foams were observed as a powder, thus losing their initial porosity and being seen as smooth surfaces. 

The evaluation of the crystallinity of the foams only revealed slight differences among them, mostly showing amorphous material with not well-defined Bragg reflections (Figure 2d). The broad reflection present in all foams is characteristic of polyphenols, such as lignin and tannins [42,43]. This broad reflection is less pronounced in the alkali lignin, increasing in intensity when analyzing the alkali lignin-based rigid foam. The variation in intensity can be attributed to the polymerization reaction with furfuryl alcohol and p-TSA acid, which might originate in a less amorphous structure when polymerized. 

From an adsorption viewpoint, the kraft lignin addition to the TF foam was observed to decrease its performance, while the alkali lignin addition improved the adsorption performance of the material (Figure 3a). This way, the AF foam prepared with 0.9 g of p-TSA was selected, regardless of not fully self-blowing, since it was the most promising foam for dye adsorption (Figure 3b) and chosen for further characterization.

As mentioned, the electron micrographs of the ground AF foam revealed particles with a mostly smooth nonporous surface, which apparently became denser and homogeneously covered by the adsorbed MB, similar to the observation taken by Budnyak et al. (Figure 4) [4]. 

After MB adsorption, two small peaks arise, suggesting the presence of new crystalline domains. MB has been studied regarding its adsorption and aggregation at different surfaces, including clay, mica, glass, quartz, metals, graphite, etc., with various orientations being proposed (i.e., from the flat deposition of MB to tilted adsorption and organization) [44]. Despite XRD being significantly less sensitive in scenarios of nonuniform adsorption, such as at low loadings like in the present work, which can result in misleading conclusions, the adsorption of MB and formation of ordered MB structures at the foam surface sounds reasonable. This is also accompanied by a decrease in the intensity of the broad reflection at 20°, which might be due to the incorporation of water into the rigid foam lattice (partial swelling), changing the intermolecular interactions of the foam. In addition, the subsequent drying process may also contribute to the observed changes in the XRD intensity [45,46]. 

Tannins and lignin have functional groups in common and this can be easily confirmed by FTIR. For instance, the band at 3400 cm^−1^ is attributed to OH stretching, and the band at 2930 cm^−1^ is assigned to the CH stretching in aromatic compounds (Figure 5) [47,48]. Nonetheless, discriminative peaks were identified in both kraft and alkali lignins (Figure 5a(I),c(I)). The aromatic skeleton vibrations at 1600, 1515, and 1426 cm^−1^, together with the CH deformation and the aromatic ring vibration at 1462 cm^−1^, are common for all lignins. Moreover, the guaiacyl unit can be identified by the presence of varying intensity peaks at 1269 cm^−1^ and 1140 cm^−1^ related to the G ring and C=O stretch, respectively, by the CH plane deformation at 1140 cm^−1^, and the CH out-of-plane vibrations at 854 and 817 cm^−1^ (Table 2) [47,49,50]. In respect to the tannin (Figure 5b(I)), a C=O carbonyl stretching mode was detected at 1732 cm^−1^, at 1605 cm^−1^ the C=C stretching from aromatic rings, and 1390 and 1330 cm^−1^ revealed the C-O stretching and the C-O-H deformation of phenols. Together with a peak at 1200 cm^−1^, assigned to the bending of C-OH and the symmetric stretching from C-O, all the vibrations mentioned above are characteristic of tannins (Table 2) [48]. In all the prepared rigid foams, a decrease in the OH band around 3400 cm^−1^ was verified, suggesting that the condensation of lignin or tannin with furfuryl alcohol might have occurred at the free position on the aromatic ring of the phenol source [21,22,29]. Moreover, an overall decrease in intensity was observed for most peaks, which should be a consequence of the complex structure and the intermolecular rearrangements after the acid-catalyzed polymerization to obtain the rigid foams [22]. 

Once MB is adsorbed, it is of notice the increase at 1600 cm^−1^ for all foams. The appearance and/or intensity increase of characteristic peaks from MB (1390 and 1330 cm^−1^) are not identifiable in the kraft lignin foam but visible in the FTIR spectra of the tannin and the alkali lignin-based rigid foams, in agreement with the highest adsorption of dye from each foam [4].

### 3.2. Methylene Blue Adsorption Kinetics

The adsorption kinetics for MB with different initial concentrations (i.e., 5, 10, 20, and 50 mg L^−1^) are shown in Figure 6. It can be observed that the adsorption capacity is dependent on the foam composition. The observed kinetics are generally characterized by an initial high rate, followed by a slower rate until an equilibrium is reached. The irregular morphology and heterogeneity of the rigid foams make adsorption a complex process. As a smaller number of active sites become available, adsorption becomes progressively slower. Ultimately, the system reaches equilibrium and remains unchanged when all possible adsorption sites are saturated, driving the MB in excess to remain in solution [4,10]. 

Depending on its rate of adsorption, it is possible to rationalize the suitability of a material for larger scale applications. The kinetic experiments (Figure 6) demonstrate a fast adsorption process in which all the foams seem to reach equilibrium after approximately 4 h for a maximum initial concentration of 50 mg L^−1^ of MB. Among the three tested foams, KF was observed to be the least effective for MB removal. Up to 48 h in contact with a 50 mg L^−1^ MB solution at neutral pH, KF removed only close to 10 mg of MB per gram of foam, while TF foam reached close to 40 mg g^−1^. On the other hand, the AF was able to remove all the MB. This shows that MB has a higher affinity for the alkali lignin than for the tannin and, lastly, for the kraft lignin [4,10]. 

In the kinetic study, pseudo first and second-order equations were applied. Both formalisms similarly fit the data, though the highest determination coefficients (r^2^) are obtained from the pseudo-second-order fits. Overall, the r^2^ decreases with increasing MB concentration, decreasing the fit quality, though better expressed by a second-order equation. The decrease is proportional to the increase in the saturation of the material from the molecules of dye; the closer to saturation, the slower the adsorption rate. Moreover, it is also dependent on the stage of adsorption due to its logarithmic behavior [51].

The reduction in the kinetic constants shows that the rate of adsorption decreases with increasing MB concentration (Table 3). The decrease in the PSO kinetic constant (*k*_2_) trend is inversely proportional to the concentration of dye, as its decrease is greater for the more efficient adsorbent and suggests that other phenomena affect the adsorption kinetics. The AF foam *k*_2_ was reduced to 0.65% for 50 mg L^−1^ of MB with respect to 5 mg L^−1^ of MB, i.e., from 84 ± 6 (g/mg.min) to 0.55 ± 0.07 (g/mg.min), the TF foam *k*_2_ decreased to 1.67% and the KF foam to 40.36%. Overall, this shows that the adsorption rate is dependent on the concentration of dye as it decreases with an increasing concentration of dye, most likely due to the progressive saturation of available bonding sites, which reduces the encounter probability between MB molecules and the available binding sites of the adsorbent [38,52]. 

### 3.3. Methylene Blue Adsorption in “Equilibrium”

The maximum adsorption capacity for each foam was assessed through adsorption experiments in equilibrium conditions (24 h) using different concentrations of MB. In agreement with the kinetic data, equilibrium experiments (Figure 7) showed that the KF foam was the least effective for MB adsorption, followed by the TF foam. On the other hand, the AF foam was found to be the most effective. When increasing the MB concentration up to 250 mg L^−^^1^, it can be observed that the maximum adsorbed amount does not reach a steady state, oscillating between 90 and 100 mg of adsorbed MB per gram of foam. These results are supported by the pseudo-steady-state hypothesis, where the rates of adsorption and desorption are equivalent [37,53]. As an assumption characteristic of multilayer adsorption, Langmuir and BET isotherm models were used (Figure 7a). Both Langmuir and BET models present very similar fits; however, the Langmuir isotherm model better describes the adsorption process. Given this, the MB adsorption on the prepared rigid foams is usually described as a monolayer adsorption [34,37]. The Langmuir model might be a better fit when the surface of the adsorbent is saturated; nonetheless, it only represents one possible layer of adsorbate. When both models are applied for lower concentrations, i.e., up to an initial MB concentration of 100 mg L^−1^, the Langmuir model has the lowest determination coefficient compared with Freundlich and BET (Figure 7b). While the Freundlich isotherm describes a nonlinear adsorption process, meaning that adsorption is not evenly distributed on the surface of the adsorbent, the BET isotherm shows the presence of multilayers on the adsorbate [34,37,54]. The produced rigid foams are rich in several functional groups, containing flavonoid moieties and active -OH, able to interact with MB mainly via electrostatic interactions, hydrogen bonding, and π-π stacking [37]. Considering this, the data suggest that the MB adsorption by the AF foam can be described as a heterogeneous multilayer process. 

The D-R model (Equation (8)) can be applied to estimate the mean free energy (kJ mol^−^^1^) and determine whether the adsorption is mainly governed by physical (i.e., *E* < 8 kJ mol^−^^1^) or chemical processes (i.e., 8 < *E* < 16 kJ mol^−^^1^) [54]. Interestingly, the calculated mean free energy up to 250 mg L^−^^1^ of MB is 11.29 kJ mol^−^^1^, thus suggesting that the adsorption is mainly a chemical driven process [34]. However, if the model is applied in a range below the concentration where the multilayer should be formed (*q*_ads_ < 25 mg g^−^^1^), the calculated mean free energy is 6.04 kJ mol^−^^1^, thus being the adsorption mainly governed by a physical related process. Reinforcing the proposed mechanism, mainly governed by noncovalent bonds as electrostatic interactions, hydrogen bonding, and π-π stacking, the higher energy value might be associated with a stronger binding between the adsorbate and the adsorbent due to multiple interactions between the identified functional groups. Hydrogen bonding and stacking interactions are known to occur in molecules, such as cellulose, thus supporting the hypothesis [55]. 

### 3.4. pH Influence on MB Adsorption

The influence of pH on MB adsorption by the prepared foams was studied by initially inferring the point of zero charge of the adsorbents. This allowed us to confirm the acidic nature of the foams and that all the prepared adsorbents possess a negative net charge above approximately pH 3, being the most negatively charged at pH 10 (Figure 8a). The minimum adsorption capacity for all foams in an MB solution of 50 mg L^−1^ was verified at pH 2. As pH increases, the removal percentage (%*R*) exponentially increases for the KF and TF foams. Regarding the AF foam, it was able to completely remove MB from the solution from pH 4 up to pH 10 (Figure 8b). In this case, the exponential increase in adsorption caused by raising pH was not visible. As the AF foam adsorption capacity in neutral conditions reaches approximately 100 mg L^−1^, the acidic pH effects might not be strong enough to be noticed in such a concentration below its maximum adsorption capacity. However, when increasing the MB solution concentration up to 250 mg L^−1^, the AF foam was able to remove 88% of the MB present in the solution at pH 10 (Figure 8c), representing a removal capacity of 220 mg of MB per g of foam. In such conditions, the AF foam demonstrates a good capacity for MB removal compared with similar foams. For instance, Sepperer et al. prepared tannin-lignin-based furan foams with a maximum MB removal capacity of 73.8 mg g^−1^, the tannin rigid foams prepared by Sánchez-Martín et al. display an MB removal capacity of 250 mg g^−1^ [10,41]. The pH can affect both the protonation state of the dye and the chromophores at the adsorbent surface, i.e., at low pH, the high amount of hydronium will compete with the cationic dye for the adsorption sites, decreasing MB adsorption. On the other hand, in alkaline conditions, the adsorbent surface is expected to be mostly negatively charged, and thus, the MB adsorption is enhanced [39,51]. 

The AF foam should be more efficient than tannin and kraft lignin-based foams due to its high number of HO^−^ as can be confirmed by its alkaline pH (10.5), while tannin has a pH close to 6, and kraft lignin between 2.5 and 4.5. The highest surface negative charges present in the alkali lignin-based foam, originating from HO^−^ deprotonation, could be responsible for the increased affinity towards the cationic MB compared with the kraft lignin-based foam [56]. In addition, commercial tannin is composed of a minimum of 65% (*w*/*w*) of polyphenols, which can justify its poorer performance compared with both tested lignins, where impurities do not reach 4% (*w*/*w*).

### 3.5. Adsorbent Reusability and Recycling

The adsorbent recycling assays showed it is possible to reuse the AF foam for MB adsorption after removing the adsorbed MB with an EtOH solution at pH 2 (Figure 9). Such solvent was chosen due to the high affinity of MB with EtOH in synergy with the low pH effect, previously discussed. The desorption efficiency is higher than in acetone or HCl 0.1 M. The higher desorption capacity in acid conditions confirms that the adsorption mechanism is mainly governed by electrostatic interactions [4,53]. In a 50 mg L^−^^1^ MB solution, the AF foam completely adsorbed the dye in the first cycle. However, the removal efficiency decreased to roughly half for the second cycle and beyond (i.e., 23 ± 3 mg g^−^^1^), coinciding with the desorption capacity of 26 ± 1 mg g^−^^1^. This effect suggests that, for this higher concentration (50 mg L^−1^), some reactive sites might be unavailable due to an incomplete removal of the dye [38,39]. It is noticed that until the last cycle, the adsorption and desorption efficiencies are equivalent. This might be due to the interplay between multiple interactions, such as hydrogen bonding and possible stacking interactions between MB and the rigid foam particles, as the suggested multilayer adsorption occurring at 50 mg L^−^^1^ of MB hinders the full desorption of the strongly adsorbed molecules. Supporting this, the same effect was not verified when the AF foam was exposed to 10 and 20 mg L^−^^1^ of the MB solution, as in both situations, it was possible to remove most of the dye; all the previously occupied reactive sites were again free for a new adsorption cycle. 

### 3.6. Thermal Behaviour

Lignin has been used as a flame retardant additive for polymer-based materials, being moderately stable at elevated temperatures due to its aromatic structure [57,58]. Thermal insulation and fire resistance tests are important features for the application of materials, such as foams in construction, appliances, transportation, and furniture [59]. Replacing fossil-based materials with biobased flame retardants can be an attractive alternative for the creation of more sustainable, high-performance polymeric materials [60]. In this regard, the thermal stability tests show that all the prepared rigid foams present similar thermal properties. The observed mass loss with increasing temperature is similar, reaching slightly over 50% of the initial mass when exposed up to 1200 °C (Figure 10a), showing improved thermal properties compared with similar materials [29]. TGA analysis exhibits several weight loss peaks, which can be attributed to the evaporation of residual blowing agent and adsorbed water between 25 and 150 °C; the degradation of polymer chains forming smaller molecules from 200 to 500 °C, and above 450 °C the pyrolysis of lignin and tannin is most likely anticipated [43,61].

The foam’s thermal stability was further evaluated by exposing it to a butane flame for 5 min (Figure 11). As can be observed, the AF foam did not produce any smoke or contribute to increasing the flame. After 5 min, and as soon as the butane source was extinguished, there was no combustion on the rigid foam, showing it was not flammable. Moreover, it took approximately 20 s to dissipate the heat, losing all the red glow. The fire retardancy of the AF rigid foam might be due to the polyfuranic oligomers formed by furfuryl alcohol self-condensation and its condensation with alkali lignin [26]. Overall, the data suggests these materials might be suitable for applications where flame retardant behavior is of utmost importance, such as with insulation materials. 

## 4. Conclusions

Lignin-furan foams were produced by completely replacing the tannin from a standard tannin-based furan rigid foam. As widely available and considered a residue from the paper industry, lignin can be considered an attractive polyphenol source for producing such materials. The rigid foams prepared from tannin or lignin showed similar thermal properties; however, they greatly differed in terms of MB adsorption. Two types of lignin were compared, and the alkali lignin was shown to be the most promising for dye adsorption, removing up to 220 mg of MB per gram of foam at pH 10. Adding to this, the adsorption mechanism was accessed by studying the process kinetics and equilibrium conditions, applying several isotherms. From this study, it was possible to characterize the adsorption process as nonlinear multilayer adsorption, most likely mainly governed by noncovalent bonds such as hydrogen bonding, electrostatic, and π-π interactions. The produced materials showed the possibility of being recycled for several adsorption-desorption cycles, though the procedure seems to be strongly dependent on the concentration of dye. The biobased-produced rigid foams also showed remarkable flame resistance properties, losing roughly half of their mass up to 1200 °C, not being flammable, and losing the red glow 20 s after being exposed for 5 min to a butane flame. This work demonstrates how a biobased residue (lignin) can be valorized and become an attractive platform to safely produce a multitask material suitable for dye adsorption, which, eventually, can be given a new end-of-life as a flame retardant with improved flame retardant capacity.

## Figures and Tables

**Figure 1 polymers-16-03315-f001:**
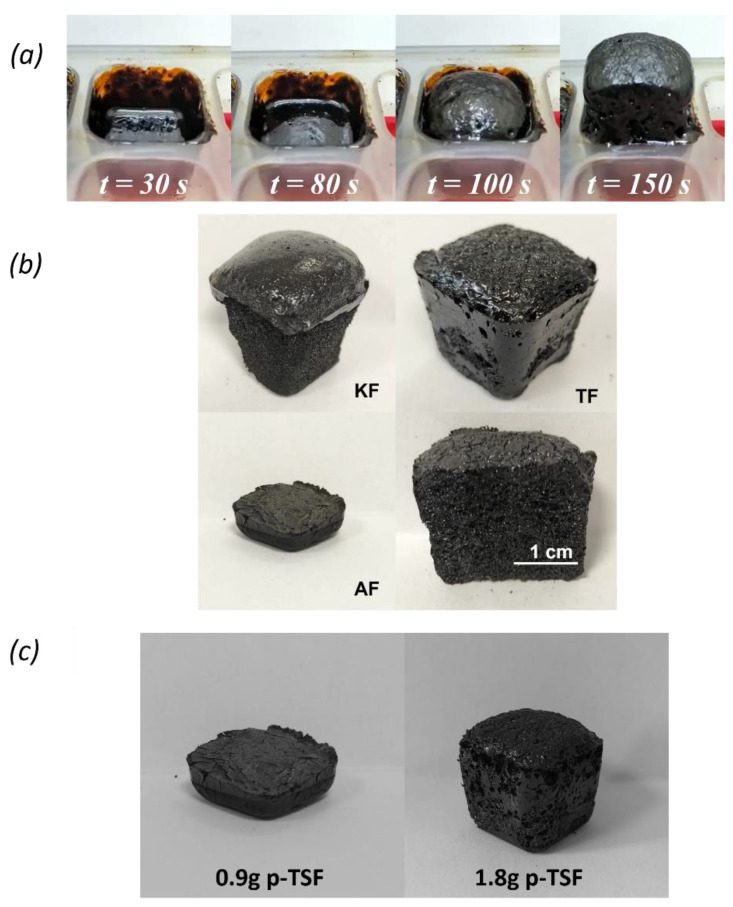
(**a**) Time development of a typical AF foam prepared with 0.9 g or 1.8 g of p-TSA acid; (**b**) Images of the prepared foams after unmolding, and cross section from the TF foam; (**c**) AF foam prepared using the standard procedure by adding 0.9 g of p-TSA, without the capacity to self-blow (**left**), in contrast to the same formulation when the amount of p-TSA was doubled, allowing the foam to self-blow (**right**) (KF—kraft-furan, AF—alkali-furan, and TF—tannin-furan foam).

**Figure 2 polymers-16-03315-f002:**
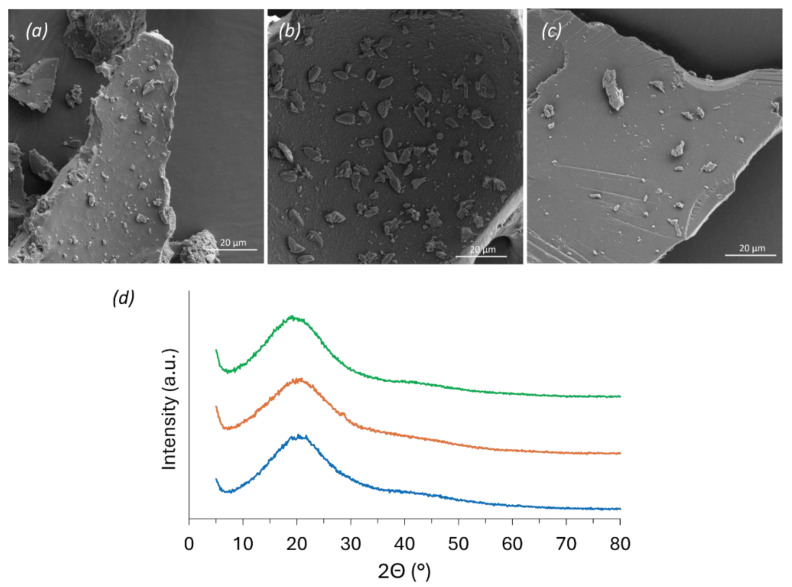
Scanning electron micrographs of KF (**a**), TF (**b**), and AF (**c**) foams; (**d**) XRD patterns for KF (green), TF (orange), and AF (blue) rigid foams.

**Figure 3 polymers-16-03315-f003:**
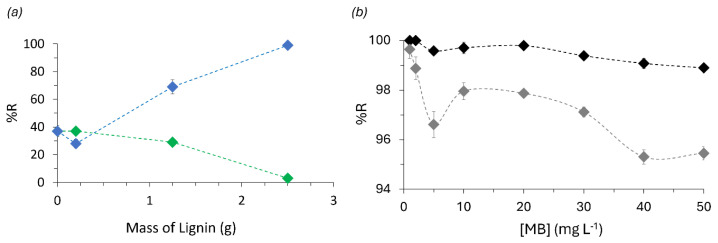
(**a**) MB removal efficiency of tannin-furan foams with increasing concentration of kraft (green) and alkali (blue) lignins; (**b**) AF foam MB removal after 24 h, by adding the standard 0.9 g (black) or 1.8 g (grey) of p-TSA acid during foam preparation.

**Figure 4 polymers-16-03315-f004:**
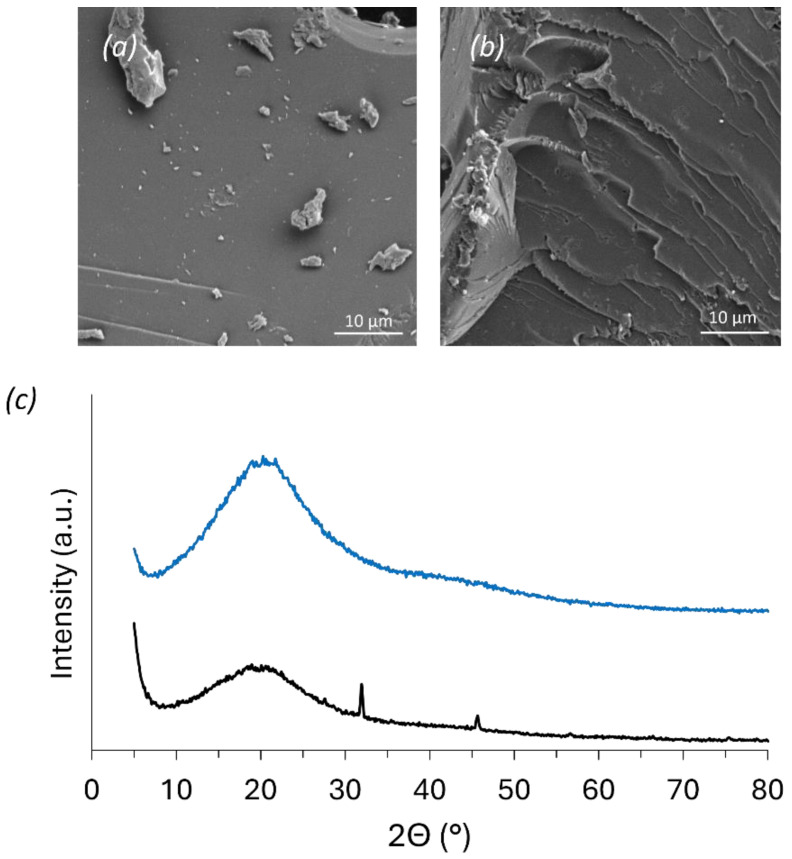
Scanning electron micrographs of grounded AF foam before (**a**) and after (**b**) MB adsorption; (**c**) XRD patterns of AF foam before (blue) and after (grey) MB adsorption.

**Figure 5 polymers-16-03315-f005:**
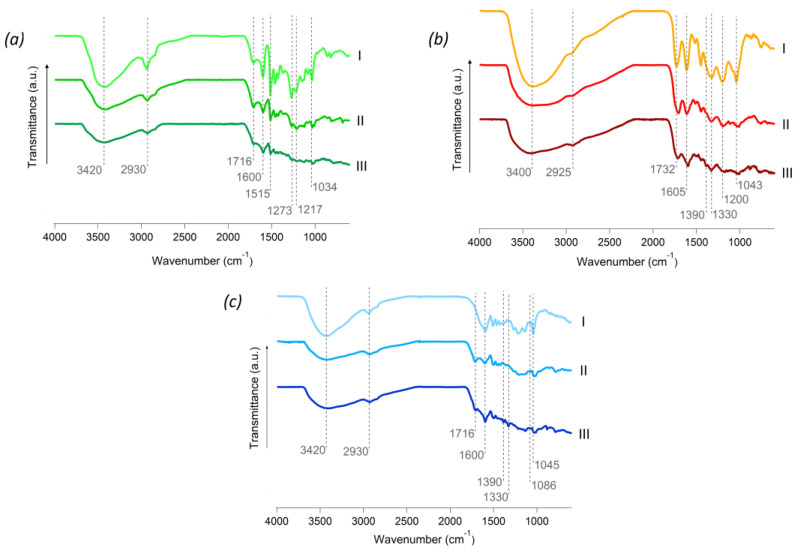
FTIR spectra for (**a**) kraft lignin (I), KF (II), and KF after MB adsorption (III); (**b**) tannin (I) and TF, before (II) and after (III) MB adsorption, (**c**) alkali lignin (I) and AF before (II) and after MB adsorption (III).

**Figure 6 polymers-16-03315-f006:**
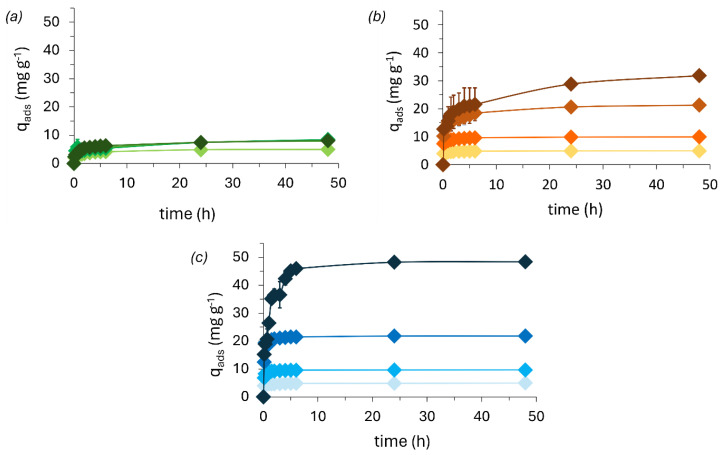
Methylene blue adsorption from (**a**) KF (green), (**b**) TF (orange), and (**c**) AF (blue) foams on the course of 48 h at 20 °C (MB initial concentrations of 5, 10, 20, and 50 mg L^−1^, increasing from lighter to darker tones.

**Figure 7 polymers-16-03315-f007:**
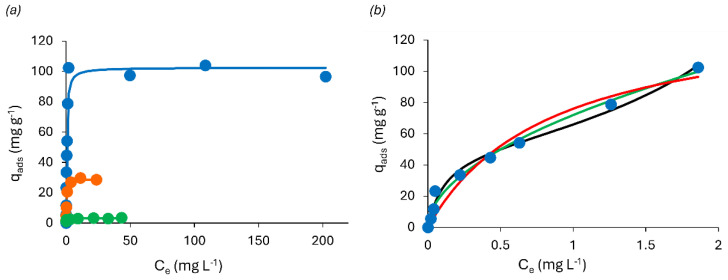
(**a**) Methylene blue adsorption for KF (green), TF (orange), and AF (blue) foams after 24 h, with the respective BET (green, orange, and blue lines) isotherm model fits, for a maximum of 250 mg L^−1^ of MB. Here, the BET model is reduced to the Langmuir isotherm (K_S_~0 Lmol^−1^) (Equation (5)); (**b**) Langmuir (red), Freundlich (green), and BET (black) model fit were applied to an initial concentration of up to 100 mg L^−1^ of MB for the AF foam.

**Figure 8 polymers-16-03315-f008:**
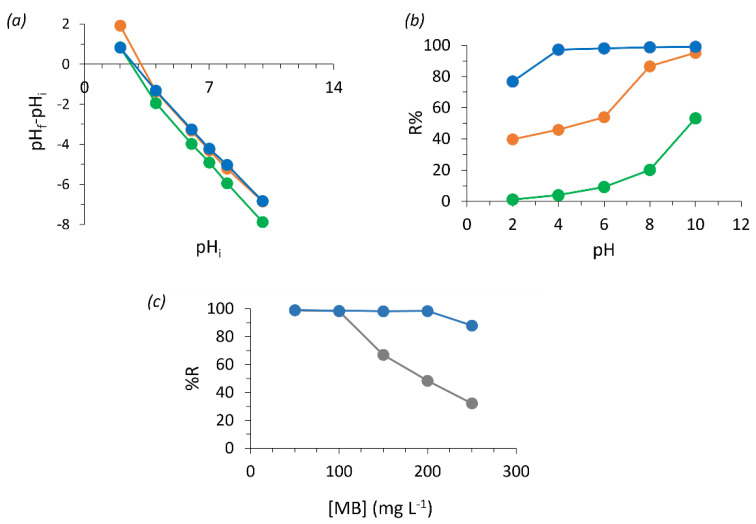
(**a**) Point of zero charge; (**b**) influence of each rigid foam on pH variation at 50 mg L^−1^ of MB, for TF (orange), KF (green), and AF (blue); and (**c**) MB adsorption on the AF rigid foam at pH 7 (grey) and 10 (blue).

**Figure 9 polymers-16-03315-f009:**
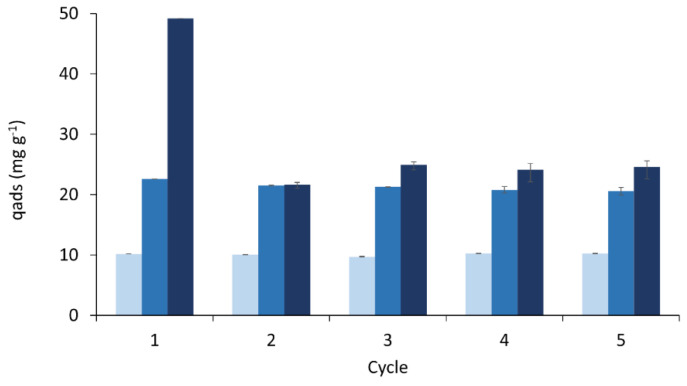
Desorption experiments during five cycles for the AF rigid foam exposed to 10 (light grey), 20 (grey), and 50 (black) mg L^−1^ of MB in a solution of EtOH at pH 2 for 24 h.

**Figure 10 polymers-16-03315-f010:**
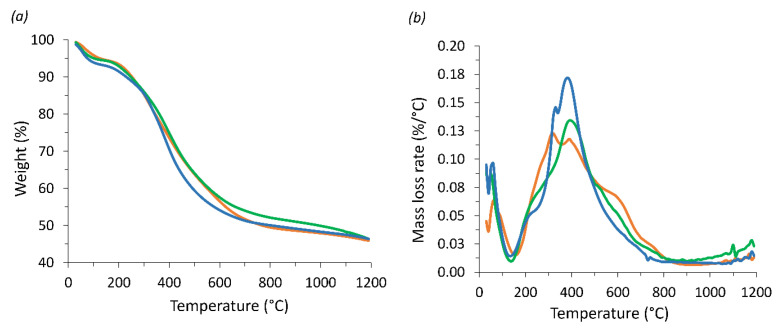
Thermogravimetric analysis of the prepared rigid foams (**a**), AF (blue), TF (orange), KF (green), and its derivative (**b**).

**Figure 11 polymers-16-03315-f011:**
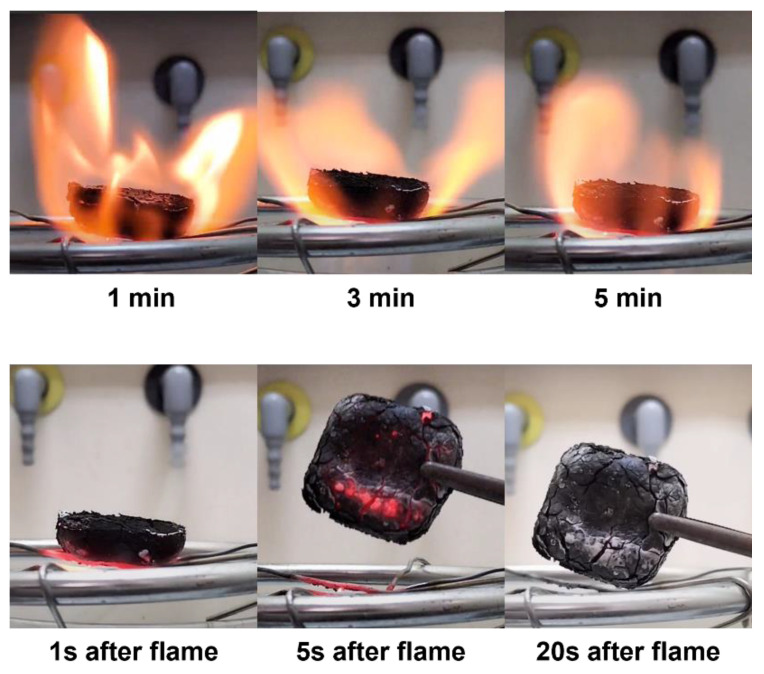
Flammability test of the AF foam under direct contact with the flame for up to 5 min (top row) and visual appearance of the foam after shutting down the gas flow (bottom row).

**Table 1 polymers-16-03315-t001:** Composition of all the prepared foams, showing how the amount of the phenol source was changed, including the mixtures of tannin and lignin.

*Foam*	*Tannin (*g*)*	*Kraft Lignin (*g*)*	*Alkali Lignin (*g*)*	*Distiled Water (*mL*)*	*Furturyl Alcohol (*mL*)*	*n-Pentane (*mL*)*	*p-Toluenesulfonic Acid (*g*)*
*Tannin (TF)*	2.4	0	0	1	1.4	0.6	0.9
	2.5	0	0	1	1.4	0.6	0.9
	2.6	0	0	1	1.4	0.6	0.9
	2.8	0	0	1	1.4	0.6	0.9
	3.0	0	0	1	1.4	0.6	0.9
*Tannin +* *kraft lignin*	2.4	0.1	0	1	1.4	0.6	0.9
	2.4	0.2	0	1	1.4	0.6	0.9
	2.4	0.5	0	1	1.4	0.6	0.9
	1.25	1.25	0	1	1.4	0.6	0.9
*Kraft lignin* *(KF)*	0	2.4	0	1	1.4	0.6	0.9
	0	2.5	0	1	1.4	0.6	0.9
	0	2.6	0	1	1.4	0.6	0.9
	0	2.8	0	1	1.4	0.6	0.9
	0	3.0	0	1	1.4	0.6	0.9
*Tannin +* *alkali lignin*	2.4	0	0.1	1	1.4	0.6	0.9
	2.4	0	0.2	1	1.4	0.6	0.9
	2.4	0	0.5	1	1.4	0.6	0.9
	1.25	0	1.25	1	1.4	0.6	0.9
*Alkali lignin* *(AF)*	0	0	2.4	1	1.4	0.6	0.9
	0	0	2.5	1	1.4	0.6	0.9
	0	0	2.5	1	1.4	0.6	1.8
	0	0	2.6	1	1.4	0.6	0.9
	0	0	2.8	1	1.4	0.6	0.9
	0	0	3.0	1	1.4	0.6	0.9

**Table 2 polymers-16-03315-t002:** FTIR peak assignment and main vibrational modes of developed foams and respective phenol source.

Sample	Wavenumber (cm^−1^)	Vibration
* **MB** *	1600	aromatic ring vibration
	1390	−CH3 symetric deformation
	1330	C=N bond
* **Tannin and TF** *	3400	OH stretching
	2925	CH stretching aromatic compounds
	1716, 1732	stretch C=O, carbonyl stretching
	1605	C=C stretching aromatic rings
	1520	stretching C−C and C−O
	1450	C−OH stretching
	1390	C−O stretching phenols
	1330	C−O−H deformation of phenols
	1200	C−OH bending
	1115	R−O−R′ stretching (ether) for condensed tannins
	1043	symmetric stretching C−O
	758	ring deformation
* **Kraft and KF** *	3420	OH stretching
	2930	CH stretching
	1716	carbonyl stretching
	1600	C=O stretching (conjugated), aromatic ring vibration
	1515	C=O stretching from aromatic rings
	1460	aromatic ring vibrations
	1430	aromatic skeletal vibrations with C−H in plane deformation
	1369	phenolic OH and aliphatic C−H in methyl groups
	1273	C=0 and ring stretching
	1217	C−C, C−O and C=O stretching
	1145	aromatic C−H in-plane deformation from guaiacyl ring,
	1086	C−O deformation in secondary alcohols and aliphatic esters
	1034	C−O stretching, C−O deformation of secondary alcohols
	856	CH in plane deformation
	820	CH out of plane vibrations
	685	ring deformation, aromatic C−H out of plane bending
* **Alkali and AF** *	3420	OH stretching
	2930	CH stretching
	1715	carbonyl stretching
	1600	C=O stretching (conjugated), aromatic ring vibration
	1510	C=O stretching from aromatic rings
	1460	aromatic ring vibrations
	1430	aromatic skeletal vibrations with C−H in plane deformation
	1390	−CH3 symetric deformation
	1330	C=N bond
	1271	C=0 and ring stretching
	1217	C−C, C−O and C=O stretching
	1145	aromatic C−H in-plane deformation from guaiacyl ring,
	1045	C−O stretching, C−O deformation of secondary alcohols
	883	CH in plane deformation
	785	CH out of plane vibrations

**Table 3 polymers-16-03315-t003:** Pseudo-first and second-order kinetics fit parameters, using Equations (3) and (4), respectively.

			Pseudo-First Order			Pseudo-Second Order		
Foam	MB (mg L^−1^)	*q_e_ exp* (mg g^−1^)	*q_e_ calc* (mg g^−1^)	*K*_1_ × 10^2^ (1 min^−1^)	*R* ^2^	*q_e_ calc* (mg g^−1^)	*K*₂ × 10^3^ (g mg^−1^ min^−1^)	*R* ^2^
TF	5	4.88 (±0.06)	4.68 (±0.06)	16.19 (±2.16)	0.98	4.80 (±0.03)	75.3 (±8.63)	0.995
	10	9.83 (±0.12)	9.18 (±0.17)	14.38 (±2.39)	0.96	9.49 (±0.11)	29.28 (±4.69)	0.986
	20	20.60 (±0.86)	17.46 (±0.70)	6.17 (±1.48)	0.85	18.84 (±0.58)	4.39 (±1.03)	0.939
	50	28.81 (±0.56)	23.14 (±1.77)	2.62 (±0.89)	0.67	25.95 (±1.80)	1.26 (±0.50)	0.815
KF	5	4.87 (±0.01)	4.17 (±0.21)	2.76 (±0.63)	0.84	4.55 (±0.17)	8.82 (±1.97)	0.938
	10	7.40 (±0.46)	6.66 (±0.63)	1.01 (±0.32)	0.74	7.48 (±0.58)	1.85 (±0.71)	0.868
	20	7.54 (±0.39)	5.82 (±0.46)	3.09 (±1.16)	0.63	6.57 (±0.47)	5.46 (±2.28)	0.794
	50	7.41 (±0.03)	6.59 (±0.34)	1.94 (±0.39)	0.87	7.32 (±0.27)	3.56 (±0.71)	0.952
AF	5	4.94 (±0.06)	4.77 (±0.04)	16.29 (±1.31)	0.99	4.88 (±0.02)	84.36 (±6.45)	0.998
	10	9.83 (±0.21)	9.61 (±0.07)	13.01 (±0.80)	0.99	9.88 (±0.03)	29.16 (±1.13)	0.999
	20	21.51 (±0.29)	20.90 (±0.21)	9.75 (±0.68)	0.99	21.71 (±0.20)	8.38 (±0.82)	0.99
	50	48.70 (±0.46)	46.24 (±1.48)	1.83 (±0.23)	0.95	50.28 (±1.14)	0.55 (±0.07)	0.98

## Data Availability

The original contributions presented in the study are included in the article, further inquiries can be directed to the corresponding author.

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
