# Peer review of "Lignin-Furanic Rigid Foams: Enhanced Methylene Blue Removal Capacity, Recyclability, and Flame Retardancy"

_polymers, 2024, doi:10.3390/polym16233315_

Round 1
Reviewer 1 Report
Comments and Suggestions for Authors
The authors have studied lignin-furan foams which were produced by completely replacing the tanning from a standard tannin-based furan rigid foam. This study illustrates how a bio-based residue, lignin, can be effectively valorized, turning it into a versatile material for dye adsorption. Ultimately, this material can be repurposed to serve as a flame retardant, offering enhanced flame-retardant properties.
All the experiments were properly designed. The manuscript is well written, however I would recommend a minor revision before accepting the manuscript.
- Wrong date in the footer of the template - 2022
- Lack of line numbering, which makes it difficult to review the work
- Abstract- Introduction: The abstract starts by stating that the key application of the studied foams is their insulating use. In the introduction, however, the focus is mainly on wastewater treatment, which might initially be misleading. The introduction and the abstract should be appropriately revised.
-Paragraph - 2.2:
A reference to a source describing the standard procedure should be included.
How were the samples stored until analysis?
-Table 1:
Information on which variants were selected for the studies and a detailed explanation of the reason should be included in the sample preparation description. Why were the TF + KF and TF + AF variants not used?
Indicate which variant serves as the control
-Figure 6: The graphics would be more readable if titles with the sample names were added.
Reviewer 2 Report
Comments and Suggestions for Authors
The manuscript describes the enhanced methylene blue removal capacity, recyclability, and flame retardancy using lignin-furanic rigid foams. The overall quality of manuscript needs to improved prior to publication.
1.Abstract - Revise the abstract. An abstract should provide short introduction, method, main findings with numerical values, short discussion and short conclusion.
2.Introduction
-Abundance of literature review/researches have reported on the polyphenol-based rigid foam. Provide novelty and gap of analysis to support the works. Compare and contrast with other literature.
-Problem statements need to be linked with national/international agenda. i.e. SDG, green technology, etc ... addition of related statistics.
-Last paragraph for aim and objectives were not clear. Be specific rather than too lengthy.
3.Materials and method
-Provide citation for all methods used
-Section 2.3 Method? or Physical, chemical and morphology characterization?
-Revise subsection title for 2.3.4
-Provide statistical methods to support results
4.Results and discussion
-Re-structure the overall section, should link with methods.
i.e. Rigid foam synthesis and adsorption dependence from lignin type and concentration (why the authors mixed SEM, FTIR, XRD, etc..?) Sub-section for each of the analysis. Re-write and sync with list of methods!
-3.1 compare and contrast the produced foam structure with literature.
-Scanning electron micrographs: label the figures with the characteristics mentioned in text i.e. rough, amorphous, etc.
-Compare and contrast SEM, XRD results with other literature.
-section 3.2: As a smaller number of active sites becomes available, adsorption becomes progressively slower... explain the mechanism clearly
-Compare and contrast Methylene blue adsorption with other literature
-Overall, this shows that the adsorption rate is dependent on the concentration of dye; it decreases with an increasing concentration of dye... Explain mechanism
-Figure 7, check x-axis and change the position.
-Provide standard deviation for all tables/figures.
-How high number of HO will make the AF foam should be more efficient than tannin and kraft lignin foams?
-Support TGA results with comparison of findings from other literature since more literature have discussed the similar subjects
5. More updated references within 5 years time 2024, 2023, 2022, 2021, 2020
Round 2
Reviewer 2 Report
Comments and Suggestions for Authors
Highlight the changes/correction in the manuscript using different colour. Answer the comment by reviewers specifically i.e. which line? paragraph ...etc rather than answer in general.
Round 3
Reviewer 2 Report
Comments and Suggestions for Authors
Acceptable correction